# Carbon-Supported Fe-Based Catalyst for Thermal-Catalytic CO_2_ Hydrogenation into C_2+_ Alcohols: The Effect of Carbon Support Porosity on Catalytic Performance

**DOI:** 10.3390/molecules29194628

**Published:** 2024-09-29

**Authors:** Yongjie Chen, Lei Jiang, Simin Lin, Pei Dong, Xiaoli Fu, Yang Wang, Qiang Liu, Mingbo Wu

**Affiliations:** 1State Key Laboratory of Heavy Oil Processing, College of New Energy, China University of Petroleum (East China), Qingdao 266580, China; s22150082@s.upc.edu.cn (Y.C.); z22150006@s.upc.edu.cn (L.J.); z21150039@s.upc.edu.cn (S.L.); 2Shandong Energy Group Co., Ltd., Jinan 250014, China; fuxiaoli@shandong-energy.com (X.F.); sdqliu@sohu.com (Q.L.)

**Keywords:** CO_2_ hydrogenation, Fe-based catalysts, porous carbon supports, C_2+_ alcohols synthesis

## Abstract

Carbon materials supported Fe-based catalysts possess great potential for the thermal-catalytic hydrogenation of CO_2_ into valuable chemicals, such as alkenes and oxygenates, due to the excellent active sites’ accessibility, appropriate interaction between the active site and carbon support, as well as the excellent capacities in C-O bond activation and C-C bond coupling. Even though tremendous progress has been made to boost the CO_2_ hydrogenation performance of carbon-supported Fe-based catalysts, e.g., additives modification, the choice of different carbon materials (graphene or carbon nanotubes), electronic property tailoring, etc., the effect of carbon support porosity on the evolution of Fe-based active sites and the corresponding catalytic performance has been rarely investigated. Herein, a series of porous carbon samples with different porosities are obtained by the K_2_CO_3_ activation of petroleum pitch under different temperatures. Fe-based active sites and the alkali promoter Na are anchored on the porous carbon to study the effect of carbon support porosity on the physicochemical properties of Fe-based active sites and CO_2_ hydrogenation performance. Multiple characterizations clarify that the bigger meso/macro-pores in the carbon support are beneficial for the formation of the Fe_5_C_2_ crystal phase for C-C bond coupling, therefore boosting the synthesis of C_2+_ chemicals, especially C_2+_ alcohols (C_2+_OH), while the limited micro-pores are unfavorable for C_2+_ chemicals synthesis owing to the sluggish crystal phase evolution and reactants’ inaccessibility. We wish our work could enrich the horizon for the rational design of highly efficient carbon-supported Fe-based catalysts.

## 1. Introduction

The resource utilization of CO_2_ can alleviate the pressure of the environmental effects of CO_2_ to a certain extent and, at the same time, provide a new green and sustainable pathway for high value-added chemical synthesis. The conversion of CO_2_ into valuable chemicals, such as olefins, aromatics, gasoline, and alcohol, through thermocatalytic hydrogenation technology is an efficient means that stands out among various CO_2_ utilization technologies due to the high conversion efficiency and promising industrial application [1,2,3,4,5,6,7,8,9,10,11,12]. Among these chemicals, C_2+_ alcohols (C_2+_OH) have received the attention of many scientists due to their wide range of uses. Most C_2+_ alcohols, including ethanol with a specific energy of 8.3 kWh/kg and an energy density of 6.7 kWh/L, hold significant economic value and an energy density comparable to gasoline (12.9 kWh/kg and 9.5 kWh/L) [13]. Many countries have used ethanol as a fuel additive or solvent for chemical products. Some C_2_-C_5_ alcohols can be used directly as a transportation fuel or blended with gasoline to increase the octane rating, thereby improving engine performance [14]. Furthermore, C_2+_ alcohols serve as essential raw materials for the production of various products, including plastics, plasticizers, and pharmaceuticals. However, the lack of highly efficient catalysts is still the bottleneck that limits the application of CO_2_ hydrogenation into C_2+_OH technology at a large scale.

Significant progress has been made in the study of CO_2_-to-C_2+_OH catalysts in recent years, mainly noble metal-based catalysts represented by Rh-based and transition metal-based catalysts represented by Co-, Cu-, and Mo-based. However, such catalysts are suffering from some inherent problems. Although the noble metal catalysts deliver high selectivity for C_2+_OH, their expensive catalyst costs and low CO_2_ conversion rates have hindered their industrialization. Transition metal-based catalysts, which are relatively cost-effective, generally suffer from low C_2+_OH selectivity, and the catalytic stability needs to be further improved [15]. Many scientists are gradually studying Fe-based catalysts because of their simplicity, ease of obtaining, low price, and lack of environmental pollution. However, block Fe-based catalysts undergo significant nanoparticle agglomeration and particle size growth during the reaction process, leading to mechanical decomposition and stability loss [16]. Therefore, Fe-based species are usually immobilized on porous supports to prolong the stability. Due to the strong interaction between the metal and the support, the general silica and alumina supports are prone to irreversibly forming inactive substances during the reaction [17]. Nanostructured carbon materials are desirable for dispersing Fe-based nanoparticles due to the appropriate metal–support interaction, high surface area, tunable texture property, and controlled surface chemistry [18,19,20,21,22]. However, most of the current research is directly loading active metal onto carbon supports [23,24,25,26,27], and there is still a great challenge for the exploration of the domain-limiting effect of pore structure on the reaction properties.

This work loaded the Na-modified Fe-based species onto the carbon supports with different pore structures that were prepared by the K_2_CO_3_ activation of petroleum asphalt at different temperatures. The porous carbon support effectively controls the size and distribution of the Fe-based nanoparticles, in which carbon supports with larger meso/macro-pores could facilitate the sufficient penetration of the Fe-based component into the pore structure and improve the dispersion. Furthermore, the support porosity influences the differential adsorption of H_2_ and CO_2_ molecules on the catalyst surface, with the CO_2_-rich and H_2_-poor environments near the Fe-based species promoting the evolution of the Fe_5_C_2_ phase as the active site, thereby enhancing CO_2_ hydrogenation performance. This result reveals the influence of carbon support porosity on the catalytic performance and provides an important guideline for future in-depth investigations into carbon-supported Fe-based catalysts for thermal-catalytic CO_2_ hydrogenation.

## 2. Results and Discussion

### 2.1. Structural Characterization of Catalysts

Figure 1a illustrates the preparation process of the carbon-supported Fe-based catalysts. Initially, petroleum asphalt underwent pyrolysis via the one-step activation method using K_2_CO_3_ at varying temperatures under an N_2_ atmosphere. The resulting carbon supports were denoted as MCx (x represents the pyrolysis temperature, x = 700/800/900/1000 °C). Subsequently, the Fe-based active sites were loaded onto a carbon support by the impregnation method followed by carbonization treatment at 550 °C for 3 h under an N_2_ atmosphere and further impregnation with the Na promoter to obtain the catalyst named NaFe/MCx.

The N_2_ adsorption–desorption test was performed on petroleum asphalt-derived porous carbon MCx to explore the effect of K_2_CO_3_ activation temperature on pore structure (Table 1). MC700 and MC1000 delivered the lowest and highest N_2_ adsorption–desorption capacity, respectively (Figure 1b,c and Table 1). For MC700, the micropores dominated the pore structure with a 926.45 m^2^ g^−1^ specific surface area of micropores. The specific surface area of mesopores accounted for only 4% of the total. Obviously, the percentage of mesopores in MCx increased with the increase in pyrolysis temperature. The specific surface area of the mesopores in MC1000 increased to 910.74 m^2^ g^−1^, constituting 55% of the total specific surface area. The increasing mesopores’ specific surface area indicated that the pore structure of MCx, especially the mesoporous property, can be modulated by adjusting the pyrolysis temperature. Despite the high surface area of microporous-dominant materials, their CO_2_ trapping ability and kinetic performance may be unsatisfactory due to the insufficient diffusion of CO_2_ molecules into the core of micropores [28]. Meso/macro-pores, conversely, possess a larger pore volume, exhibit excellent CO_2_ trapping capacity under high pressure, facilitate faster mass transfer [29], and provide sufficient sites for anchoring Fe-based active sites, thus enhancing accessibility to catalytically active sites [30].

Transmission electron microscope (TEM) characterization was performed on the fresh NaFe/MCx catalysts to observe the distribution of Fe-based particles on the carbon support (Figure 2). Based on the findings from Figure 1 and Table 1, the specific surface area of the MC700 catalyst was the smallest, with micropores dominating the pore structure. This prevents most of the Fe-based particles from entering the pores of the carbon support, resulting in large particle sizes accumulated outside the carbon support with the largest average diameter of 15.37 nm (Figure 2a). With the pyrolysis temperature increased, the average pore size of the carbon support was enlarged. Therefore, the Fe-based nanoparticles were more easily dispersed into the pore structure, causing the reduced average diameter of the Fe-based active sites (Figure 2b,c). A TEM image of NaFe/MC1000 revealed uniformly dispersed Fe-based nanoparticles without an apparent accumulation on the carbon support. The Fe-based nanoparticles with an average diameter of 8.69 nm were well dispersed on the carbon supports and encapsulated in the pores of MC1000 (Figure 2d). The uniform distribution of Fe-based active sites inside the pores of carbon supports could enhance the provision of catalytic active centers that are easily accessible to gaseous reactants, thus improving the catalytic performance of thermal-catalytic CO_2_ hydrogenation.

X-ray diffraction (XRD) characterization elucidated the crystal phase of NaFe/MCx catalysts (Figure 3a). The fresh NaFe/MCx catalyst was primarily composed of the Fe_3_O_4_ phase. With the increase in the activation temperature, the dispersion of the Fe-based component increased, accompanied by a decrease in Fe_3_O_4_ diffraction peak intensity, which is consistent with the TEM characterization results. Fe_3_O_4_ is the main active site of the reverse water–gas shift (RWGS, CO_2_ + H_2_ → CO + H_2_O) reaction, which dissociates CO_2_ into CO for the subsequent Fischer–Tropsch synthesis [31]. As the reaction proceeds, Fe sites undergo gradual carburization, transforming from Fe_3_O_4_ to Fe-based carbide phases. H_2_-temperature programmed reduction (H_2_-TPR) recorded the reduction behavior of the catalysts. As shown in Figure 3b, the H_2_-TPR curves showed the multi-step reduction processes of Fe_3_O_4_ in catalysts [32,33]. The exposure degree of Fe-based active sites under the reducing atmosphere determines the reduction behavior of the catalyst. Benefitting from the smallest particle size, highest intra-pore dispersion, and substantial pore volume, NaFe/MC1000 exhibited superior reduction behavior among all samples, with a slight shift in the reduction peak towards lower temperatures. Conversely, larger Fe-based nanoparticles entering the pore structure diminished Fe-based active sites’ ability to contact H_2_, resulting in a wider reduction process of NaFe/MC900 and NaFe/MC800. The largest catalyst particle size of NaFe/MC700 hindered the reduction behavior but primarily stacked outside pore structures, maintaining relatively strong H_2_ contact, hence exhibiting a significantly shorter reduction interval [34,35,36].

Figure 3c,d represent the CO-temperature programmed reduction (CO-TPR) profiles of the NaFe/MCx catalysts and the XRD patterns of the catalysts after CO-TPR, respectively. The peaks that appeared in the CO-TPR curves represented the consumption of CO, indicating that CO can reduce the Fe-based nanoparticles for carburization and form Fe-based carbide compounds during the CO-TPR process. The appearance of Fe_3_C-related peaks in the XRD patterns of the CO-TPR catalysts also confirmed this conclusion. Notably, the NaFe/MC1000 catalyst delivered the lowest CO-TPR temperature, implying that the smallest Fe-based nanoparticles endowed by the mesoporous carbon support were easier to be reduced and carbonized by CO molecules. However, for the NaFe/MC700 catalyst, no obvious CO-TPR peaks were detected due to the difficulty of the reduction and carbonization of the largest Fe-based nanoparticles that stacked outside the micropores.

### 2.2. Catalytic Performance

The Fe-based component served as the active site, facilitating the dissociation of CO_2_, carbon chain growth, and the insertion of oxygen-containing intermediates. Consequently, the Na/MC1000 sample was inactive due to the absence of these crucial sites (Table 2). Na acts as a promoter, increasing the surface basicity of Fe catalysts, facilitating the formation of iron carbide phases, and inhibiting the over-hydrogenation process by strengthening the desorption of desirable products (especially alkenes) [37,38,39]. As a result, Fe/MC1000 catalysts without Na doping exhibited lower C_2+_ alcohols selectivity and higher selectivity of CH_4_ and paraffins (Table 2). Although the contents of Fe and Na were consistent among different catalysts, they showed completely different catalytic performance, as shown in Table 2 and Figure 4. NaFe/MC700 exhibited a CO_2_ conversion of 6.1% and the highest CH_4_ selectivity of 90.9%, devoid of valuable oxygenates in the product. As the pyrolysis temperature of the carbon support increased, the CO_2_ conversion rate and C_2+_OH selectivity gradually rose, while CH_4_ selectivity declined. NaFe/MC1000 exhibited the highest CO_2_ conversion (22.8%) and selectivity towards C_2+_OH (22.6%), alongside the lowest CH_4_ selectivity (22.5%). Furthermore, over a 48 h testing period, the CO_2_ conversion and product distribution of NaFe/MC1000 remained stable without being impacted by carbon deposition or pore blockage (Table 2 and Figure 4c,d). We believe that the pore structure of the catalysts endowed by different pyrolysis temperatures plays a decisive role in the reaction performance, and the increased proportion of the mesopores in the catalysts will improve the conversion of CO_2_ and the selectivity of C_2+_OH.

Figure 4b exhibits the trend of the CO_2_ conversion rate of NaFe/MCx. The observed gradual increase in CO_2_ conversion over time can be attributed to the induction period required by the Fe-based catalyst to achieve a stable active state (Figure 4b). This process involves the remodeling of the catalyst surface, during which the Fe-based catalyst undergoes carburization during the reaction and gradually forms surface active sites represented by Fe_3_O_4_ and Fe_5_C_2_ to adsorb and activate CO_2_ molecules [40]. The CO_2_ conversion rate of NaFe/MC700 markedly declined after 6 h of reaction initiation. The CO_2_ conversion rate of NaFe/MC700 markedly declined after 6 h of reaction initiation. This trend can be attributed to the predominantly microporous structure of NaFe/MC700. The microporous structure limits the accessibility and evolution of the Fe-based active sites, resulting in suboptimal catalytic performance. Additionally, catalysts with a primarily microporous structure are more prone to significant deactivation due to the serious carbon deposition phenomenon [41].

### 2.3. Influence of Catalyst Pore Structure on Catalytic Performance

Figure 5 reflects the morphology of Fe-based particles in the spent NaFe/MCx catalysts. The TEM images revealed no significant aggregation of Fe-based particles on the carbon supports, indicating that the proper interaction between carbon supports and metal particles effectively prevents agglomeration during the reaction. The particle sizes increased to varying degrees after the CO_2_ hydrogenation reaction, indicating distinct evolution patterns of Fe species driven by different pore-structured supports. Specifically, for NaFe/MC1000 catalysts (Figure 5d), the prevalence of mesopores promotes significant Fe species evolution during reaction, as evidenced by the obvious growth of Fe-based particle size.

N_2_ adsorption–desorption tests were conducted on fresh catalysts to explore the pore structure of the NaFe/MCx catalysts (Figure 6a,b). NaFe/MC700 and NaFe/MC900 exhibited the lowest and highest N_2_ adsorption–desorption capacities, respectively. NaFe/MC700 had the lowest specific surface area of 710.77 m^2^ g^−1^, with micropores dominating the pore structure. The specific surface area of NaFe/MCx was gradually increased with the increase in the activation temperature, but exhibited a reduction when carbon support pyrolysis reached 1000 °C. This reduction could be attributed to the graphitic carbon layer formation in the NaFe/MC1000 catalyst as the activation temperature reached 1000 °C. Additionally, the adsorption hysteresis loop appeared in the N_2_ adsorption–desorption curve of the NaFe/MC1000 (Figure 6a), which indicates that the pores retained after pyrolysis are mainly mesoporous [41]. Notably, the mesoporous specific surface area of NaFe/MC1000 reached the highest value of 357.49 m^2^ g^−1^ (Table 3 and Figure 6b). The changes in the pore structure of the carbon support during pyrolysis were related to the activation of K_2_CO_3_. During the chemical activation process, a higher activation temperature intensified the K_2_CO_3_ activation effect, enhancing etching and pore creation in carbon material [42].

The adsorption capacity of carbon-supported Fe-based catalysts to the feedstock gas components (CO_2_ and H_2_) significantly influences their reaction performance. Accordingly, physical adsorption–desorption tests for CO_2_ and H_2_ were conducted. The results indicated that the adsorption capacities for both CO_2_ and H_2_ increased with the pressure rising to 3 bar (Figure 6c,d). Notably, the CO_2_ adsorption capacity consistently surpassed that for H2, attributable to the inherent properties of Fe-based catalysts. Incorporating transition metal oxides like Cu and Ni into the carbon support enhances CO_2_ adsorption capability [43,44,45,46]. Similarly, Fe-based species on the carbon support also enhance CO_2_ adsorption capability significantly. NaFe/MC700 exhibited the lowest CO_2_ adsorption capacity, while its adsorption capacity for H_2_ was higher (Figure 6c,d). H_2_ is more readily diffused into the micropore-dominated structure of NaFe/MC700, creating an H_2_-rich environment near the active sites during reaction and leading to the over-hydrogenation phenomenon. This behavior corresponds well with the high CH_4_ selectivity observed in NaFe/MC700 (Table 2). Conversely, the mesoporous NaFe/MC1000 catalyst demonstrated the highest H_2_ physisorption and substantial CO_2_ physisorption, and the larger meso/macro pores in the carbon support facilitated the effective diffusion of both gases, matching the rates of CO_2_ activation and hydrogenation to boost the reaction performance (Figure 6c,d). Consequently, the CH_4_ selectivity was reduced, and the selectivity for C_2+_OH increased to 22.6% (Table 2), underscoring the role of an enhanced mesoporous structure in promoting the formation of C_2+_OH.

The temperature-programmed desorption mass spectrometry (TPD-MS) of CO_2_/H_2_ was conducted on NaFe/MCx catalysts to examine the chemisorption behavior of the feed gas components. As the pyrolysis temperature increased, the position of the CO_2_ desorption peak shifted toward lower temperatures (Figure 7a), with the NaFe@MC1000 catalyst exhibiting the lowest CO_2_ desorption temperature due to the higher mesopore content. The H_2_ desorption peak for the NaFe/MC700 catalyst shifted to higher temperatures and intensified (Figure 7b), indicating strong H_2_ adsorption on the catalyst surface, likely facilitating H_2_-rich environments conducive to the over-hydrogenation of -CO* intermediates and predominant production of CH_4_ (Table 2). As the pyrolysis temperature increased, the catalyst’s H_2_ adsorption capacity decreased, establishing a catalytic interface with moderate hydrogenation potential that effectively suppressed undesirable hydrogenation reactions and enhanced carburization effects.

Figure 7c,d show the XRD and X-ray Photoelectron Spectroscopy (XPS) profiles of the spent NaFe/MCx catalysts. For the spent catalysts, Fe_3_O_4_ was the mainly oxide phase, while Fe_3_C and Fe_5_C_2_ were mainly the Fe-based carbide phases. The XPS characterization of the spent NaFe/MCx catalysts revealed the formation of Fe-C bonds in all catalysts, indicating the presence of Fe-based carbide compounds, consistent with the XRD results. There was almost no formation of the Fe-based carbide phase in the predominantly microporous NaFe/MC700 catalyst, and the signals of the Fe-based carbide phase in the spent catalysts became more pronounced with the increase in mesopores. As is known, the active phases of Fe_3_O_4_ and Fe-based carbide make outstanding contributions to the thermal-catalytic CO_2_ hydrogenation reaction, in which CO_2_ molecules are converted to -CO* intermediates by the Fe_3_O_4_ active site via the RWGS reaction. Subsequently, -CO* carburizes the Fe_3_O_4_ active phase to carbides dominated by the Fe_3_C and Fe_5_C_2_ phases with H_2_-assisted action [47,48], while the Fe-based carbides dissociate or non-dissociatively activate the -CO* intermediates to -CH_x_* or -CH_y_O* (x = 1, 2, or 3, and y = 0, 1, or 2), respectively, and further C-C coupling with subsequent hydrogenation steps occur to finalize the synthesis of C_2+_OH [3,49].

## 3. Materials and Methods

### 3.1. Materials

Petroleum asphalt was supplied by the Sinopec Jiu Jiang Company (Jiu Jiang, China). Potassium carbonate (K_2_CO_3_, AR) and sodium carbonate (Na_2_CO_3_, AR) were obtained from Sinopharm Chemical Reagent Co., Ltd (Shanghai, China). Ferric nitrate nonahydrate (Fe(NO_3_)_3_·9H_2_O, AR) was purchased from Shanghai Macklin Biochemical Co., Ltd. (Shanghai, China). All reagents were used directly without further purification.

### 3.2. Catalyst Preparation

The carbon supports were prepared by a one-step heat-treatment method using K_2_CO_3_ chemical activation. First, 1 g of petroleum asphalt and 4 g of K_2_CO_3_ were ground and mixed thoroughly, then heated up to 700/800/900/1000 °C for 2 h in a tube furnace under an N_2_ atmosphere at a heating rate of 5 °C·min^−1^ to obtain the black samples. Subsequently, the samples were washed with deionized water for 12 h at 70 °C and filtered three times to remove unreacted salts, ensuring that no other elements in the catalyst interfered with the experiments. Then, the samples were dried in an oven at 60 °C for 12 h to obtain carbon supports named MC700/800/900/1000, respectively. In the next step, the carbon supports were impregnated with a 15% Fe(NO_3_)_3_.9H_2_O solution, dried at 70 °C for 12 h, and then pyrolyzed at 550 °C under an N_2_ atmosphere for 3 h. The Na-modified carbon-supported Fe-based catalysts, named NaFe/MCx (x represents the temperature of the calcination, x = 700/800/900/1000), were then fabricated by the impregnation method under the conditions of 3% Na_2_CO_3_, a 70 °C thermal treatment, and a 12 h reaction time. Finally, all catalysts were pressed, crushed, and sieved to 20–40 meshes for the CO_2_ hydrogenation catalytic performance test.

### 3.3. Catalyst Characterization

The X-ray diffraction (XRD) patterns of the catalysts’ powder were characterized with a Rigaku RINT 2400 X-ray diffractometer using Cu K α-radiation. Transmission electron microscopy (TEM, JEM-2100UHR, JEOL) was used to observe the morphologies of the catalysts before and after the reaction. The crystal phase structures found in the experiments were determined using the PDF–4+ (2019) crystal database and then matched using Highscore Plus software. X-ray photoelectron spectroscopy (XPS, ThermoFisher (Beijing, China), ESCALAB 250Xi) analysis of the reacted catalysts was recorded to determine the elemental composition and valence changes. The specific surface areas and pore size distributions of the catalysts were measured by a Micromeritics 3Flex 2 M P instrument. Before the measurements, the catalysts were degassed at 200 °C for 6 h. The element content of NaFe/MCx catalysts was determined by an inductively coupled plasma optical emission spectrometry (ICP-OES, Agilent ICP-720ES). The specific surface area was calculated by the Brunauer–Emmett–Teller (BET) method. The average pore size and pore volume were calculated using the Barrett–Joyner–Halenda (BJH) method. The pore size distribution was analyzed by the BJH (mesopores) and Horváth–Kawazoe (HK, micropores) methods.

At different temperatures and pressures, the gas adsorption isotherms for CO_2_ and H_2_ were determined using the volumetric method on a high-pressure gas adsorption apparatus (BSD-PH, BeiShiDe Instrument Co., Ltd., Beijing, China). The procedure involved loading approximately 0.1 g of the powder sample into the sample tube. Before initiating the gas adsorption test, the sample was placed in a heating furnace, warmed to 200 °C, and activated under vacuum conditions for 5 h. During the adsorption test, the adsorption temperature was precisely controlled using a program-controlled water bath jacket. Various pressure set points were maintained through computer program-controlled procedures.

CO/H_2_ temperature-programmed reduction (CO/H_2_-TPR) and CO_2_/H_2_-temperature-programmed desorption and mass spectra (CO_2_/H_2_-TPD-MS) were performed on a PCA-1200 instrument connected to an MS-200 mass spectrometer (Beijing Builder, Beijing, China). For CO_2_/H_2_-TPD, a 30 mg catalyst was pretreated at 150 °C for 30 min under a flow of pure Ar to remove the physically adsorbed water and organic products on the spent catalyst. Then, the sample was saturated with CO_2_/H_2_ at 50 °C for 1 h. After removing the physically adsorbed CO_2_/H_2_ by a He flow, the CO_2_/H_2_-TPD curve was collected under the He flow (30 mL min^−1^) with a heating rate of 10 °C min^−1^. The signals of the desorbed H_2_ (*m*/*z* = 2) and CO_2_ (*m*/*z* = 44) were detected by MS and a thermal conductivity detector (TCD). CO/H_2_-TPR was carried out in a stream of 10% CO/H_2_ in Ar with a heating rate of 10 °C min^−1^.

### 3.4. Catalytic Evaluation

The catalytic activity tests were performed on a continuous flow type fixed-bed reactor with an inner diameter of 6 mm. For the catalytic performance tests, 0.1 g of NaFe/MC700/800/900/1000 catalysts and 1 g of quartz sand were fixed in the middle of the reactor by quartz wool, with 2 g of quartz sand used as a spoiler above the catalyst bed. Before the reaction, the catalyst was reduced by pure H_2_ at 400 °C for 4 h. After cooling to room temperature, the reactant gas (23.75% CO_2_, 71.23% H_2_, and 5.02% Ar) was fed into the reactor until the pressure reached 5 MPa. At the same time, the temperature of the reactor was increased to 320 °C. The heavy hydrocarbons were collected by the ice trap set between the reactor and the back pressure valve, and then analyzed by an off-line gas chromatograph (GC9790II) equipped with an FID and an InerCap-5 capillary column (GL Sciences (Shanghai, China), 0.25 mm × 30 m). The exhaust gas was analyzed by two on-line gas chromatographs (Fuli 9790II, Zhejiang Fuli Analytical Instruments Co., Ltd., Taizhou, China), in which one was equipped with a TCD detector and an active charcoal column for the analysis of Ar, CO, CH_4_, and CO_2_, and the other was equipped with an FID detector and a Porapak-Q column for the analysis of light hydrocarbons. CO_2_ or CO (denoted as CO_x_, x = 1 or 2) conversion and product selectivity were calculated using the following equations.

(1) CO_2_ conversion was calculated according to the following:$${CO}_{2}\;Conversion=\frac{{CO}_{2inlet}-{CO}_{2outlet}}{{CO}_{2inlet}}\times 100\%$$
where ${CO}_{2inlet}$ and ${CO}_{2outlet}$ represent the moles of CO_2_ at the inlet and outlet, respectively.

(2) Product selectivity, the percentage of CO_2_ converted into a given product, was calculated as follows:$${Sel}_{i}=\frac{N_{i}\times n_{i}}{\sum_{1}^{i}\left(N_{i}\times n_{i}\right)}\times 100\%$$
where $N_{i}$ and $n_{i}$ represent the mole percentage and carbon number of product i.

(3) CO selectivity (CO selectivity for CO_2_ hydrogenation) was calculated according to the following:$$CO\;selectivity=\frac{{CO}_{outlet}}{{CO}_{2inlet}-{CO}_{2outlet}}\times 100\%$$

The carbon balances of the reaction data were calculated, and all were higher than 90%. Typically, the experimental data after the reaction of 24 h were used for discussion.

## 4. Conclusions

Porous carbon support Na-modified Fe-based catalysts were prepared by K_2_CO_3_ activation combined with the impregnation method using cheap and readily available petroleum asphalt as a carbon source. The fabricated catalysts were applied in the CO_2_ hydrogenation reaction to obtain high value-added C_2+_OH chemicals. Varying the activation temperature of petroleum asphalt obtained NaFe/MCx catalysts with different pore structures. The NaFe/MC1000 catalyst with the highest percentage of mesopores showed the most excellent catalytic activity. Under the reaction conditions of 320 °C, 5 MPa, and GHSV = 9000 mL g_cat_^−1^ h^−1^, the CO_2_ conversion was 22.8%, and the C_2+_ alcohol selectivity was 22.6%. The characterization results showed that the pore-limiting role of carbon support influenced the size and distribution of Fe-based nanoparticles and the adsorption pattern of reactive gases, which in turn affects the formation of the Fe_5_C_2_ active phase by carburization and, ultimately, the performance of CO_2_ hydrogenation. The catalysts with a larger mesoporous specific surface area exhibited higher CO_2_ conversion and C_2+_OH selectivity due to the preferential formation of the Fe_5_C_2_ crystalline phase for C-C coupling. We wish our work could provide guidance for the rational design of the carbon-supported Fe-based catalyst for CO_2_ hydrogenation into C_2+_OH.

## Figures and Tables

**Figure 1 molecules-29-04628-f001:**
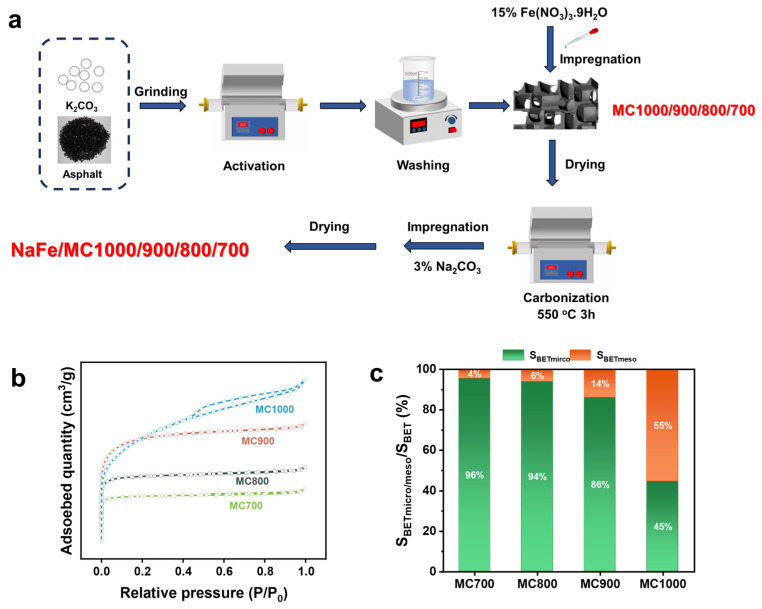
(**a**) The preparation process of the carbon-supported Fe-based catalyst, (**b**) N_2_ adsorption–desorption, and (**c**) the ratio of the microporous/mesoporous specific surface area of the porous carbon MCx.

**Figure 2 molecules-29-04628-f002:**
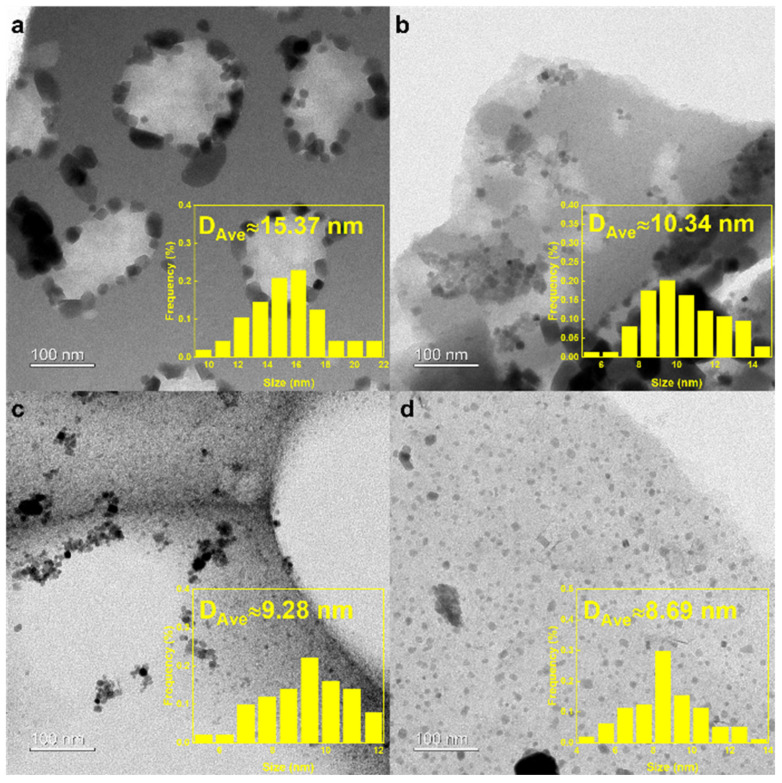
TEM images of the fresh (**a**) NaFe/MC700, (**b**) NaFe/MC800, (**c**) NaFe/MC900, and (**d**) NaFe/MC1000 with different Fe particle sizes.

**Figure 3 molecules-29-04628-f003:**
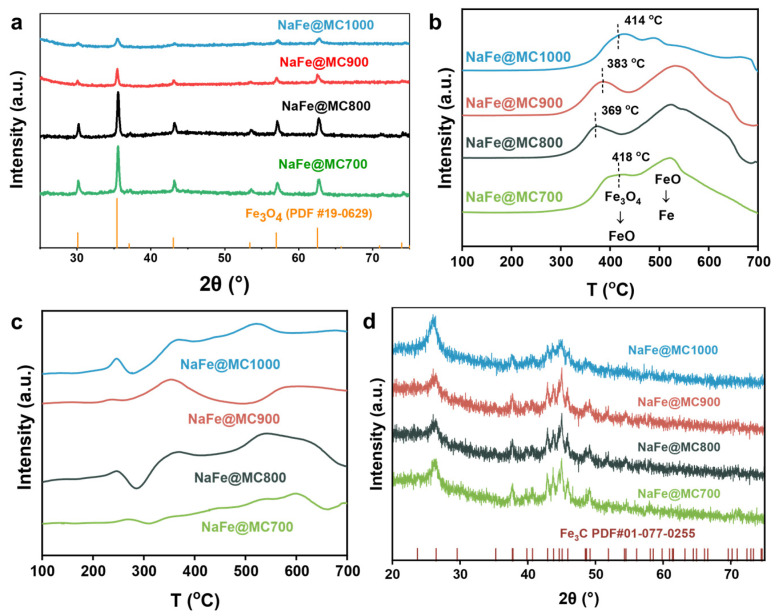
(**a**) XRD patterns and (**b**) H_2_-TPR profiles of the NaFe/MCx, (**c**) CO-TPR profiles of NaFe/MCx, and (**d**) XRD patterns of NaFe/MCx catalyst after CO-TPR.

**Figure 4 molecules-29-04628-f004:**
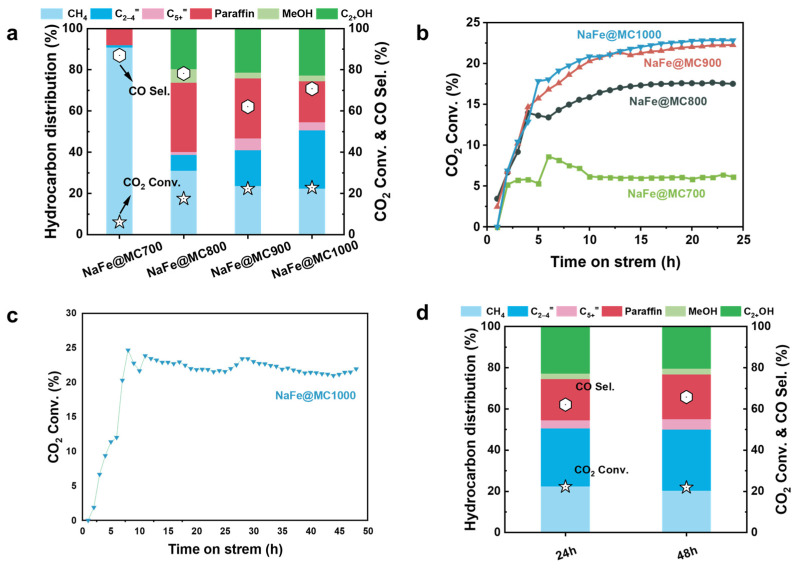
(**a**) Product distribution of the NaFe/MCx catalysts in CO_2_ hydrogenation. (The bars represent selectivity: light blue, methane; dark blue, C_2_–C_4_ olefins; light red, C_5+_ olefins; dark red, paraffinic hydrocarbons; light green, methanol; dark green, C_2+_ alcohols. The hexagonal icon represents CO selectivity and the star icon represents CO_2_ conversion.) (**b**) CO_2_ conversion rate of NaFe/MCx over time. (TOS) = 24 h. (**c**) CO_2_ conversion rate and (**d**) product distribution of NaFe/MC1000 over time. (TOS) = 48 h.

**Figure 5 molecules-29-04628-f005:**
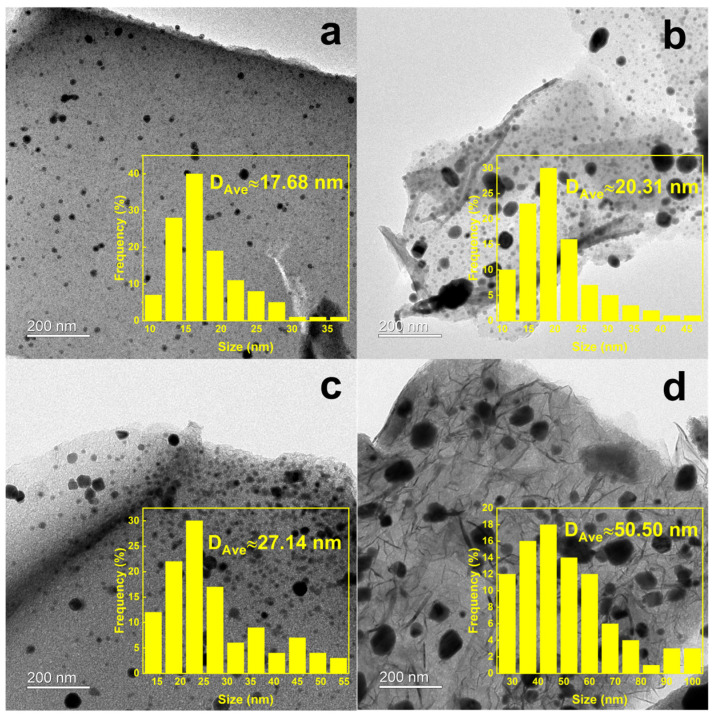
TEM images of the spent (**a**) NaFe/MC700, (**b**) NaFe/MC800, (**c**) NaFe/MC900, and (**d**) NaFe/MC1000 with different Fe particle sizes.

**Figure 6 molecules-29-04628-f006:**
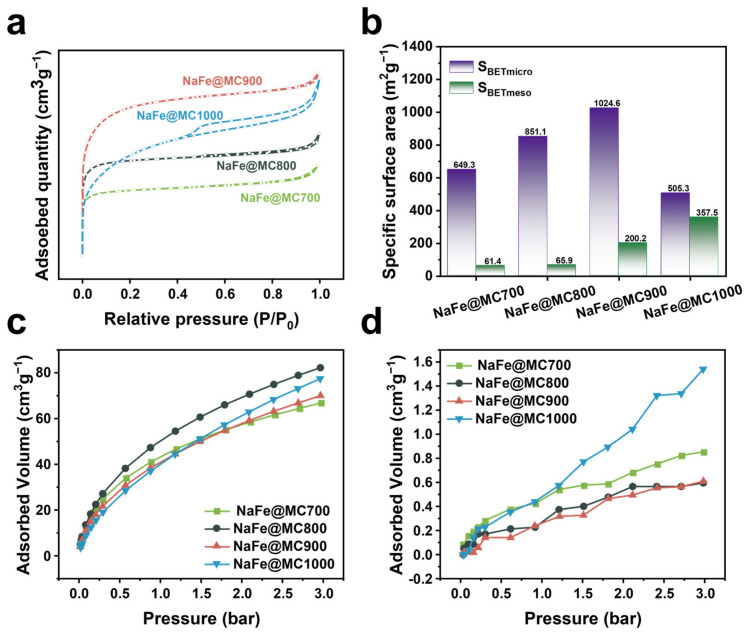
(**a**) N_2_ adsorption−desorption and (**b**) micro/meso porous specific surface area of the NaFe/MCx. (**c**) CO_2_ adsorption curves and (**d**) H_2_ adsorption curves of NaFe/MCx.

**Figure 7 molecules-29-04628-f007:**
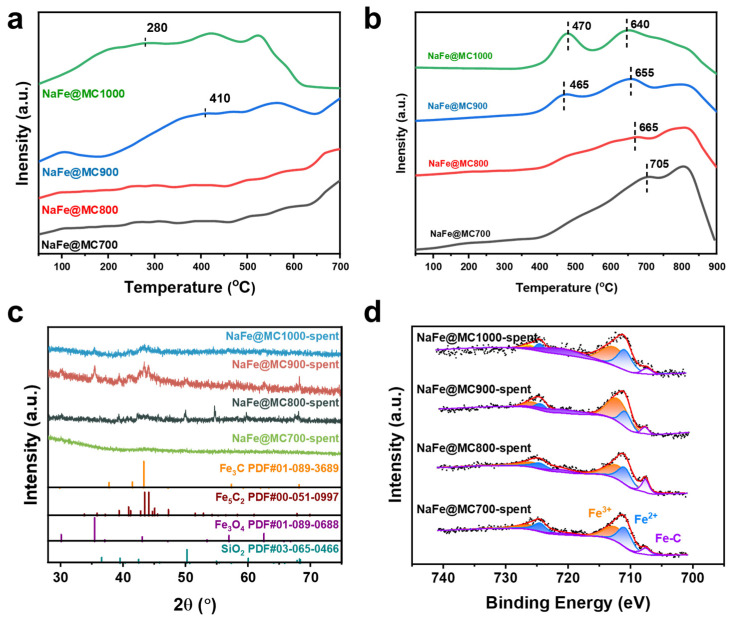
(**a**) CO_2_-TPD-MS and (**b**) H_2_-TPD-MS profiles of the NaFe/MCx, (**c**) XRD patterns, and (**d**) XPS profiles of the spent NaFe/MCx.

**Table 1 molecules-29-04628-t001:** Texture properties of MCx.

Catalyst	S_BET_ (m^2^ g^−1^)	S_BETmicro_ (m^2^ g^−1^)	S_BETmeso_ (m^2^ g^−1^)	V_micro_ (cm^3^ g^−1^)	V_meso_ (cm^3^ g^−1^)	d_size_ (nm)
MC700	969.04	926.45	42.59	0.34	0.06	1.67
MC800	1254.20	1178.86	75.34	0.44	0.09	1.70
MC900	1732.19	1492.01	240.18	0.59	0.20	1.83
MC1000	1645.13	734.38	910.74	0.25	0.68	2.54

**Table 2 molecules-29-04628-t002:** Catalytic performance of the NaFe/MCx catalysts in CO_2_ hydrogenation.

Catalyst	CO_2_ Conv. (%)	CO Sel. (%)	Hydrocarbons				MeOH	C_2+_OH	C_2+_OH STY mg g_cat_^−1^ h^−1^	Yield
			CH_4_	C_2–4_ =	C_5+_ =	Paraffin				
NaFe/MC700	6.1	87.0	90.9	1.1	0.0	8.0	0.0	0.0	0.0	0.0
NaFe/MC800	17.5	78.2	31.2	7.6	1.5	33.6	6.6	19.5	15.2	0.7
NaFe/MC900	22.3	62.1	23.8	17.3	5.7	29.2	2.8	21.3	36.2	1.8
NaFe/MC1000	22.8	70.8	22.5	28.2	3.9	20.0	2.7	22.6	30.3	1.5
Fe/MC1000	18.4	87.05	36.8	30.3	1.3	31.2	0.1	4.5	0.8	0.8
Na/MC1000	1.6	-	-	-	-	-	-	-	-	-

Reaction conditions: 320 °C, 5 MPa (23.75% CO_2_, 71.23% H_2_, and 5.02% Ar), 15 mL min^−1^, GHSV = 9000 mL gcat^−1^ h^−1^, 1 g quartz sand, and time on stream (TOS) = 24 h. Catalyst weight: 0.1 g.

**Table 3 molecules-29-04628-t003:** Texture properties of NaFe/MCx.

Catalyst	S_BET_ (m^2^ g^−1^)	S_BETmicro_ (m^2^ g^−1^)	S_BETmeso_ (m^2^ g^−1^)	V_micro_ (cm^3^ g^−1^)	V_meso_ (cm^3^ g^−1^)	d_size_ (nm)	Fe Content ^a^ (%)
NaFe/MC700	710.77	649.34	61.43	0.25	0.09	1.93	22.1
NaFe/MC800	916.97	851.07	65.90	0.33	0.10	1.80	19.9
NaFe/MC900	1224.74	1024.59	200.15	0.40	0.19	1.92	20.5
NaFe/MC1000	862.76	505.27	357.49	0.19	0.36	2.55	21.3

^a^ Element content in catalysts as characterized by ICP-AES.

## Data Availability

The raw data supporting the conclusions of this article will be made available by the authors on request.

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
