# Peer review of "Carbon-Supported Fe-Based Catalyst for Thermal-Catalytic CO2 Hydrogenation into C2+ Alcohols: The Effect of Carbon Support Porosity on Catalytic Performance"

_molecules, 2024, doi:10.3390/molecules29194628_

Round 1

Reviewer 1 Report

Comments and Suggestions for Authors

Comments on the manuscript “Carbon-supported Fe-based catalyst for thermal-catalytic CO2 hydrogenation into C2+ alcohols: The effect of carbon support porosity on catalytic performance” by, Yongjie Chen, Lei Jiang, Simin Lin, Pei Dong, Xiaoli Fu, Yang Wang, Qiang Liu, Mingbo Wu, submitted to MDPI molecules.

 The authors describe an inventive method of supporting the Fe-based catalyst particles for the conversion of CO2 with H2 to produce CH4 and certain oxygenated products such as alcohols.  The work appears to be done with care, the authors use several characterization techniques before and after catalytic testing.  The catalytic results, that is both the degree of CO2 conversion and the selectivity of products are discussed it terms of their relation to the variable conditions of catalyst synthesis.  The paper seems publishable, but after the author having given thought to the following points.

 1) p. 4  It would help the reader if the authors discussing Fig. 2 clearly stated which panel they refer to in particular point, simply using a), b), and so on.  Also, the reader may realize that the parameter DAve is the size of Fe-particle, but still it should be clearly stated in the figure caption.

 2) p. 6, line 177: Why do the authors consider CH4 as less valuable than the oxygenated products?

 3) p. 6, Figure 4b Why is the CO2 conversion increasing in time?

 4) p.7, Figure 5.  The figure is hardly comprehensible, please do increase lettering,  In the caption, please state what Dave means here.  It seems not the same as in Fig. 2.

 5) Figures 6 and 7 are unintelligible; please increase the panels and lettering.

 6) p. 11, line 345.  What is a role of this 5 % of Ar in the reaction mixture?

 7) A general question: may the author provide any hint as to the mechanism of the CO2+H2 conversion; in particular about the way that the H2 molecule may be dissociated on the catalyst surface during reaction?  Admittedly, I am just being curious; the authors may not include this point in their response if they feel no to.

 8) p. 1, line 34: Excessive use of fossil fuels results in massive CO2 emissions and serious environmental problems. For example, large amounts of CO2 emissions lead to global warming, ocean acidification, and other environmental problems, causing severe impacts on human production and life [1]. 

These two sentences do not contribute to any aspect of research presented in the manuscript and do not support any conclusion that the authors have reached.  The so called “global warming” as related to the human made CO2 emissions is a political issue, not a scientific issue.  Obviously, the authors are free to follow and to support any political option of their choice.  However, a scientific publication should not be a venue of political agitation.  Therefore I suggest the authors to remove these sentences.

Author Response

Reviewer 1

Comments and Suggestions for Authors

The authors describe an inventive method of supporting the Fe-based catalyst particles for the conversion of CO2 with H2 to produce CH4 and certain oxygenated products such as alcohols. The work appears to be done with care, the authors use several characterization techniques before and after catalytic testing. The catalytic results, that is both the degree of CO2 conversion and the selectivity of products are discussed it terms of their relation to the variable conditions of catalyst synthesis. The paper seems publishable, but after the author having given thought to the following points.

Response:

The authors thank Reviewer 1 for the valuable comments. Detailed amendments are given below.

Comments 1: p.4 It would help the reader if the authors discussing Fig. 2 clearly stated which panel they refer to in particular point, simply using a), b), and so on. Also, the reader may realize that the parameter DAve is the size of Fe-particle, but still it should be clearly stated in the figure caption.

Response 1:

Modifications have been made in the corresponding paragraphs and headings on p.4 in the revised manuscript.

Update

p.4. “This prevents most of the Fe-based particles from entering the pores of the carbon support, resulting in large particle sizes accumulated outside the carbon support with the largest average diameter of 15.37 nm (Figure 2a). With the pyrolysis temperature increased, the average pore size of carbon support enlarged. Therefore, the Fe-based nanoparticles were more easily dispersed into the pore structure, causing the reduced average diameter of the Fe-based active sites (Figure 2b, c). TEM image of NaFe/MC1000 revealed uniformly dispersed Fe-based nanoparticles without apparent accumulation on carbon support. The Fe-based nanoparticles with average diameter of 8.69 nm were well dispersed on the carbon supports and encapsulated in the pores of MC1000 (Figure 2d).”.

p.4. “Figure 2. TEM images of the fresh (a) NaFe/MC700, (b) NaFe/MC800, (c) NaFe/MC900, and (d) NaFe/MC1000 with different Fe particle sizes.”

Comments 2: p. 6, line 177: Why do the authors consider CH4 as less valuable than the oxygenated products?

Response 2:

Corresponding additions have been made to the Introduction section of the revised manuscript, and the corresponding literature has been cited in the revised manuscript.

Even though methane (CH4) is an important fuel or chemical, its market value is generally lower than that of oxygenated products such as C2+ alcohols due to its lower energy density (2.5 kWh/L as Compressed Natural Gas and 6.1 kWh/L as Liquefied Natural Gas). Furthermore, owing to the high activation energy of C-H bond in CH4, the high energy consumption for CH4 conversion into valuable chemicals limits its utilization as feedstock for chemicals synthesis. Therefore, in our opinion, the high energy density and excellent chemicals synthesis potential of C2+ alcohols guarantee their greater market value.

Update

p.1. “Most C2+ alcohols, including ethanol with a specific energy of 8.3 kWh/kg and an energy density of 6.7 kWh/L, hold significant economic value and energy density comparable to gasoline (12.9 kWh/kg and 9.5 kWh/L) [13]. Many countries have used ethanol as a fuel additive or solvent for chemical products. Some C2-C5 alcohols can be used directly as a transportation fuel or blended with gasoline to increase octane rating, thereby improving engine performance [14]. Furthermore, C2+ alcohols serve as essential raw materials for the production of various products, including plastics, plasticizers, and pharmaceuticals.

Ref. 13 and Ref. 14 have been cited in the revised manuscript.

  1. Shyama, C. M., Amitabha, D., Diptendu, R., Sandeep, D., Akhil, S. N., Long, C., Biswarup, P. (2022). Developments of the heterogeneous and homogeneous CO2 hydrogenation to value-added C2+-based hydrocarbons and oxygenated products. Coord Chem Rev. 471, 214737.
  2. Ho, T. L., Cecilia, M., Daniel, C. F., Joseph, A. S., Javier, P. R. (2017). Status and prospects in higher alcohols synthesis from syngas. Chem. Soc. Rev. 46, 1358.

Comments 3: p. 6, Figure 4b Why is the CO2 conversion increasing in time?

Response 3:

The revised manuscript discusses the relevant content and cites the relevant references.

Update

p.7. “The observed gradual increase in CO2 conversion over time can be attributed to the induction period required by the Fe-based catalyst to achieve a stable active state (Figure 4b). This process involves the remodeling of the catalyst surface, during which the Fe-based catalyst undergoes carburization during the reaction and gradually forms surface active sites represented by Fe3O4 and Fe5C2 to adsorb and activate CO2 molecules [41].”

Ref. 41 have been cited in the revised manuscript.

  1. Han, X. X., Zhao, Q., Gong, H., Wei, C., Lv, J., Wang, Y., Wang, M., Huang, S., and Ma, X. B. (2023). Interface-Induced Phase Evolution and Spatial Distribution of Fe-Based Catalysts for Fischer-Tropsch Synthesis. ACS Catal. 13, 6525-6535.

Comments 4: p.7, Figure 5. The figure is hardly comprehensible, please do increase lettering, In the caption, please state what Dave means here. It seems not the same as in Fig. 2.

Response 4:

The term DAve in Fig. 5 refers to the particle size of Fe-based nanoparticles. Fig. 2 and Fig. 5 represent TEM images of fresh and spent catalysts, respectively. These figures illustrate the differences in the evolution of Fe species on various pore structure supports. Particularly for NaFe/MC1000 catalysts, the abundance of mesopores facilitates a complete evolution of Fe species during the reaction, reflected by the growth of Fe particle size. This has been detailed in the revised manuscript. Furthermore, the lettering in Fig. 5 has been modified.

Update

p.7. “The particle sizes increased to varying degrees after CO2 hydrogenation reaction, indicating distinct evolution patterns of Fe species driven by different pore-structured supports. Specifically, for NaFe/MC1000 catalysts (Figure 5d), the prevalence of mesopores promotes significant Fe species evolution during reaction, as evidenced by the obvious growth of Fe-based particle size.”

p.4. “Figure 2. TEM images of the fresh (a) NaFe/MC700, (b) NaFe/MC800, (c) NaFe/MC900, and (d) NaFe/MC1000 with different Fe particle sizes.”

p.7.” Figure 5. TEM images of the spent (a) NaFe/MC700, (b) NaFe/MC800, (c) NaFe/MC900, and (d) NaFe/MC1000 with different Fe particle sizes.”

Comments 5: Figures 6 and 7 are unintelligible; please increase the panels and lettering.

Response 5:

In the revised manuscript, we have enhanced the panels and lettering to ensure the figures more comprehensible. Key information and trends in each panel have been highlighted, and the chart titles and legend descriptions have been refined to provide clearer explanations of the data.

Update

p.8. “The adsorption capacity of carbon-supported Fe-based catalysts to the feedstock gas components (CO2 and H2) significantly influences their reaction performance. Accordingly, physical adsorption-desorption tests for CO2 and H2 were conducted. Results indicated that the adsorption capacities for both CO2 and H2 increased with the pressure rising to 3 bar (Figure 6c, d). Notably, the CO2 adsorption capacity consistently surpassed that for H2, attributable to the inherent properties of Fe-based catalysts. Incorporating transition metal oxides like Cu and Ni into the carbon support enhances CO2 adsorption capability [44-47]. Similarly, Fe-based species on the carbon support also enhance CO2 adsorption capability significantly. NaFe/MC700 exhibited the lowest CO2 adsorption capacity, while its adsorption capacity for H2 was higher (Figure 6 c,d). H2 is more readily diffused into the micropore-dominated structure of NaFe/MC700, creating an H2-rich environment near the active sites during reaction and leading to over-hydrogenation phenomenon. This behavior corresponds well with the high CH4 selectivity observed in NaFe/MC700 (Table 2). Conversely, the mesoporous NaFe/MC1000 catalyst demonstrated the highest H2 physisorption and substantial CO2 physisorption ((Figure 6 c,d)), and the larger meso/macro pores in the carbon support facilitated effective diffusion of both gases, matching the rates of CO2 activation and hydrogenation to boost the reaction performance. Consequently, the CH4 selectivity was reduced, and the selectivity for C2+OH increased to 22.6% (Table 2), underscoring the role of an enhanced mesoporous structure in promoting the formation of C2+OH.”

p.9. “Figure 6. (a) N2 adsorption-desorption and (b) micro/meso porous specific surface area of the NaFe/MCx. (c) CO2 adsorption curves and (d) H2 adsorption curves of NaFe/MCx.”

p. 9. “The H2 desorption peak for the NaFe/MC700 catalyst shifted to higher temperatures and intensified (Figure 7b), indicating strong H2 adsorption on the catalyst surface, likely facilitating a H2-rich environments conducive to over-hydrogenation of -CO* intermediates and predominant production of CH4 (Table 2). As pyrolysis temperature increased, the catalyst's H2 adsorption capacity decreased, establishing a catalytic interface with moderate hydrogenation potential that effectively suppressed undesirable hydrogenation reactions and enhanced carburization effects.”

Comments 6: p. 11, line 345. What is a role of this 5 % of Ar in the reaction mixture?

Response 6:

In CO2 hydrogenation activity test, 5% Ar was used as an internal standard to analyze the change of the gas composition during the process and calculate the CO2 conversion based on the following equations.

CO2 conversion was calculated according to:

Where CO2inlet and CO2outlet represent moles of CO2 at the inlet and outlet, numerically equal to the division of the chromatographic TCD peak areas of CO2 and Ar.

Comments 7: A general question: may the author provide any hint as to the mechanism of the CO2+H2 conversion; in particular about the way that the H2 molecule may be dissociated on the catalyst surface during reaction?  Admittedly, I am just being curious; the authors may not include this point in their response if they feel no to.

Response 7:

Additions and relevant references have been made in the corresponding sections of the revised manuscript.

The widely accepted catalytic mechanism for CO2 hydrogenation to C2+ alcohols over Fe-based catalysts involves the initial conversion of CO2 to -CO* intermediates via the reverse water-gas shift (RWGS) reaction at Fe3O4 active sites. Subsequently, these -CO* intermediates carburize the Fe3O4 phase into a carbide phase dominated by χ-Fe5C2 with the assistance of H2. This carbide phase then activates the -CO* intermediates into -CHm* or -CHnO* intermediates (where m = 1, 2, or 3 and n = 0, 1, or 2). Further C-C coupling and hydrogenation steps lead to the synthesis of C2+ alcohols.

Essentially, the adsorption and evolutionary processes of the key reaction intermediate -CO* are closely related to the electronic structural properties of the Fe-based active site. The activation of the key reaction intermediate -CO* is achieved through the supply of electrons to the π*-anti-bonding orbitals of -CO* from the Fe-based active site. This electron supply strengthens the Fe-C bond while weakening the C-O bond. By adjusting the electronic structure properties of Fe-based active sites and optimizing the electron donation behavior of Fe atoms to the π*-antibonding orbitals of -CO* intermediates, it is possible to balance the dissociative and nondissociative processes of the C-O bond. This balance enables the rate-matched generation of -CHm* and -CHnO* intermediates and optimizes the coverage of intermediates and the energy barriers of C-C couplings, thereby stimulating the optimal performance of the catalyst for the synthesis of C2+ alcohols.

Regarding H2 dissociation on the catalyst surface, there is still a lack of detailed research in the field of CO2 hydrogenation on Fe-based catalysts. However, studies in other areas of CO2 hydrogenation suggest that H2 dissociates at metal sites, migrates to CO2-adsorbed active sites through hydrogen spillover, and participates in the hydrogenation of reaction intermediates. This mechanism is influenced significantly by the nature of the catalyst, with relevant studies cited in the revised manuscript.

Update

Ref. 48, Ref. 49 and Ref. 50 have been cited in the revised manuscript.

  1. Zhang, D., Luo, J., Wang, J., Xiao, X., Liu, Y., Qi, W., Su, D. S., and Chu, W. (2018). Ru/FeOx catalyst performance design: Highly dispersed Ru species for selective carbon dioxide hydrogenation. Chin. J. Catal. 39, 157-166.
  2. Wu, C., Shen, J., An, X., Wu, Z., Qian, S., Zhang, S., Wang, Z., Song, B., Cheng, Y., Sham, T. K., Zhang, X., Li, C., Feng, K., and He, L. (2024). Phosphorization-induced “Fence Effect” on the active hydrogen species migration enables tunable CO2 hydrogenation selectivity. ACS Catal. 14, 8592-8601.
  3. Wei, J., Ge, Q., Yao, R., Wen, Z., Fang, C., Guo, L., Xu, H., and Sun, J. (2017). Directly converting CO2 into a gasoline fuel. Nat. Commun. 8, 15174.

Comments 8: p. 1, line 34: “Excessive use of fossil fuels results in massive CO2 emissions and serious environmental problems. For example, large amounts of CO2 emissions lead to global warming, ocean acidification, and other environmental problems, causing severe impacts on human production and life [1].” 

These two sentences do not contribute to any aspect of research presented in the manuscript and do not support any conclusion that the authors have reached. The so called “global warming” as related to the human made CO2 emissions is a political issue, not a scientific issue. Obviously, the authors are free to follow and to support any political option of their choice. However, a scientific publication should not be a venue of political agitation. Therefore I suggest the authors to remove these sentences.

Response 8:

The corresponding content has been removed from the manuscript to maintain a focus on the scientific aspects of our research.

Reviewer 2 Report

Comments and Suggestions for Authors

The study investigates the impact of carbon-support porosity on the catalytic performance of a carbon-supported iron-based catalyst for thermal-catalytic CO2 hydrogenation into C2+ alcohols. Results indicate that higher catalyst porosity enhances catalytic activity. However, several points need to be addressed to strengthen the manuscript:

  1. Although the modification of Na and Fe species reduced the specific surface area of MC700/800/900/1000, it did not significantly affect the pore size. Does MC700/800/900/1000 affect the reaction?
  2. What are the actual iron contents in the samples?
  3. The article states: "With the increase in pyrolysis temperature, the specific surface area of the catalyst gradually increased, and mesopores proliferated, providing more adsorption sites for CO2 and resulting in the gradual increase of CO2 conversion during the reaction." However, the specific surface area of the most performant NaFe@MC1000 is lower than that of NaFe@MC900 and NaFe@MC800. Is there a direct relationship between CO2 adsorption or activation and specific surface area? If not, please reanalyze the data reasonably.
  4. In Figure 4b, why does the conversion rate of NaFe@700 show a stable trend from 5% to 10%? Could this be due to improper operations during the test? It is recommended to retest.
  5. As the calcination temperature increases, the particle size of iron species in the samples decreases before the reaction. From the TEM images, it can be observed that only in the NaFe@MC700 sample, iron is coated around the carrier, while at other calcination temperatures, the iron component is uniformly dispersed on the carrier. Therefore, is the naming method of the catalysts inappropriate? It cannot define catalyst structure using encapsulation.
  6. What is the effect of Na addition on catalytic activity? The results only showed a comparison between MC1000 and NaFe@MC1000. Does Fe@MC1000 perform better? The performance of Na@MC1000 and Fe@MC1000 should be provided.
  7. After the reaction, the significant increase in particle size of iron species is due to aggregation or the formation of carbides affecting pore size. However, Figure 4b shows that the catalyst remains stable after 25 hours of reaction. Is there a difference in product selectivity due to carburization or pore blockage during the reaction process? The selectivity results of the product within 0-25 hours should be reflected in the main text.

Author Response

Comments and Suggestions for Authors

The study investigates the impact of carbon-support porosity on the catalytic performance of a carbon-supported iron-based catalyst for thermal-catalytic CO2 hydrogenation into C2+ alcohols. Results indicate that higher catalyst porosity enhances catalytic activity. However, several points need to be addressed to strengthen the manuscript:

Response:

The authors thank Reviewer 2 for the valuable comments. Detailed amendments are given below.

Comments 1: Although the modification of Na and Fe species reduced the specific surface area of MC700/800/900/1000, it did not significantly affect the pore size. Does MC700/800/900/1000 affect the reaction?

Response 1:

MC700/800/900/1000 serve as supports for loading reactive metals and do not directly participate in the reaction. The main active phase for CO2 and H2 activation is the Fe-based active phase, while the supports with different pore structures influence the reactivity primarily by affecting the evolution of the Fe-based active sites during the reaction. The reactivity enhancement observed with the increase of mesopores and macropores in both the support and the Fe-based catalysts suggests that the mesoporous/macroporous structure is the main factor influencing reactivity (SBETmeso/Vmeso).

Update

p.6. “Table 2. Catalytic performance of the NaFe/MCx catalysts in CO2 hydrogenation”

Table 2. Catalytic performance of the NaFe/MCx catalysts in CO2 hydrogenation

Catalyst

CO2

Conv. (%)

CO

Sel. (%)

Hydrocarbons

MeOH

C2+OH

C2+OH STY mg gcat-1 h-1

Yield

CH4

C2-4=

C5+=

Paraffin

NaFe/MC700

6.1

87.0

90.9

1.1

0.0

8.0

0.0

0.0

0.0

0.0

NaFe/MC800

17.5

78.2

31.2

7.6

1.5

33.6

6.6

19.5

15.2

0.7

NaFe/MC900

22.3

62.1

23.8

17.3

5.7

29.2

2.8

21.3

36.2

1.8

NaFe/MC1000

22.8

70.8

22.5

28.2

3.9

20.0

2.7

22.6

30.3

1.5

Fe/MC1000

18.4

87.05

36.8

30.3

1.3

31.2

0.1

4.5

0.8

0.8

Na/MC1000

1.6

-

-

-

-

-

-

-

-

-

Reaction conditions: 320 oC, 5 MPa (23.75% CO2, 71.23% H2, and 5.02% Ar), 15 mL min-1, GHSV=9000 mL gcat-1 h-1, 1 g quartz sand, and time on stream (TOS) = 24 h. Catalyst weight: 0.1 g.

 p.5. “The Fe-based component serves as the active site, facilitating the dissociation of CO2, carbon chain growth, and the insertion of oxygen-containing intermediates. Coversely, the Na/MC1000 sample was inactive due to the absence of these crucial sites (Table 2).”

Comments 2: What are the actual iron contents in the samples?

Response 2:

The actual iron content has been provided in Table 3 of the revised manuscript.

We further measured the Fe content in the NaFe/MCx catalyst by inductive coupled plasma-atomic emission spectroscopy (ICP-AES, Table 3)

Update

p.8. “Table 3. Texture properties of NaFe/MCx.”

Table 3. Texture properties of NaFe/MCx.

Catalyst

SBET

(m2 g-1)

SBETmicro (m2 g-1)

SBETmeso (m2 g-1)

Vmicro

(cm3 g-1)

Vmeso

(cm3 g-1)

dsize

(nm)

Fe contenta

(%)

NaFe/MC700

710.77

649.34

61.43

0.25

0.09

1.93

22.1

NaFe/MC800

916.97

851.07

65.90

0.33

0.10

1.80

19.9

NaFe/MC900

1224.74

1024.59

200.15

0.40

0.19

1.92

20.5

NaFe/MC1000

862.76

505.27

357.49

0.19

0.36

2.55

21.3

aElement content in catalysts as characterized by ICP-AES.

p.11. “The element content of NaFe/MCx catalysts was determined by an inductively coupled plasma optical emission spectrometry (ICP-OES, Agilent ICP-720ES).”

Comments 3: The article states: "With the increase in pyrolysis temperature, the specific surface area of the catalyst gradually increased, and mesopores proliferated, providing more adsorption sites for CO2 and resulting in the gradual increase of CO2 conversion during the reaction." However, the specific surface area of the most performant NaFe@MC1000 is lower than that of NaFe@MC900 and NaFe@MC800. Is there a direct relationship between CO2 adsorption or activation and specific surface area? If not, please reanalyze the data reasonably.

Response 3:

A more precise description has been provided in the revised manuscript.

Although there was a decrease in SBET, NaFe/MC1000 was the catalyst with the largest mesopore specific surface area of the samples, i.e., a higher distribution of mesopores, and there was a significant increase in mesopores of both the carriers and catalysts with the increase in heat treatment temperature (SBETmeso/Vmeso). The difference in the pore structure of the support is the dominant factor affecting the CO2 hydrogenation performance of this series of catalysts. The presence of more mesopores and macropores facilitates the phase evolution of Fe-based active sites and optimizes the adsorption behaviors of the reactants during the reaction, thereby enhancing the catalytic performance.

Update

p.8. “The adsorption capacity of carbon-supported Fe-based catalysts to the feedstock gas components (CO2 and H2) significantly influences their reaction performance. Accordingly, physical adsorption-desorption tests for CO2 and H2 were conducted. Results indicated that the adsorption capacities for both CO2 and H2 increased with the pressure rising to 3 bar (Figure 6c, d). Notably, the CO2 adsorption capacity consistently surpassed that for H2, attributable to the inherent properties of Fe-based catalysts. Incorporating transition metal oxides like Cu and Ni into the carbon support enhances CO2 adsorption capability [44-47]. Similarly, Fe-based species on the carbon support also enhance CO2 adsorption capability significantly. NaFe/MC700 exhibited the lowest CO2 adsorption capacity, while its adsorption capacity for H2 was higher. H2 is more readily diffused into the micropore-dominated structure of NaFe/MC700, creating an H2-rich environment near the active sites during reaction and leading to over-hydrogenation phenomenon. This behavior corresponds well with the high CH4 selectivity observed in NaFe/MC700 (Table 2). Conversely, the mesoporous NaFe/MC1000 catalyst demonstrated the highest H2 physisorption and substantial CO2 physisorption, and the larger meso/macro pores in the carbon support facilitated effective diffusion of both gases, matching the rates of CO2 activation and hydrogenation to boost the reaction performance. Consequently, the CH4 selectivity was reduced, and the selectivity for C2+OH increased to 22.6%, underscoring the role of an enhanced mesoporous structure in promoting the formation of C2+OH.”

Comments 4: In Figure 4b, why does the conversion rate of NaFe@700 show a stable trend from 5% to 10%? Could this be due to improper operations during the test? It is recommended to retest.

Response 4:

We retested the catalyst performance of NaFe/MC700, and the results showed good reproducibility. The conversion rate displayed an initial increase followed by a decrease in CO2 conversion, consistent with the original findings. This trend can be attributed to the predominantly microporous texture of NaFe/MC700. The microporous structure affects the accessibility and evolution of active Fe sites and may lead to more pronounced deactivation due to carbon deposition. This phenomenon has been further discussed in the revised manuscript.

Update

p.7. “The CO2 conversion rate of NaFe/MC700 markedly declined after 6 hours of reaction initiation. This trend can be attributed to the predominantly microporous structure of NaFe/MC700. The microporous structure limits accessibility and evolution of the Fe-based active sites, resulting in suboptimal catalytic performance. Additionally, catalysts with a primarily microporous structure are more prone to significant deactivation due to the serious carbon deposition phenomenon [42].”

Comments 5: As the calcination temperature increases, the particle size of iron species in the samples decreases before the reaction. From the TEM images, it can be observed that only in the NaFe@MC700 sample, iron is coated around the carrier, while at other calcination temperatures, the iron component is uniformly dispersed on the carrier. Therefore, is the naming method of the catalysts inappropriate? It cannot define catalyst structure using encapsulation.

Response 5:

The naming convention for the catalysts has been standardized to the NaFe/MCx format to better reflect their structure and composition.

Comments 6: What is the effect of Na addition on catalytic activity? The results only showed a comparison between MC1000 and NaFe@MC1000. Does Fe@MC1000 perform better? The performance of Na@MC1000 and Fe@MC1000 should be provided.

Response 6:

Performance data and analyses for Fe/MC1000 and Na/MC1000 have been added to the revised manuscript, along with relevant references. Fe-based species act as active sites for CO2 hydrogenation to ethanol, and thus Na/MC1000 samples are inactive due to the lack of these active sites. Na acts as a promoter, increasing the surface basicity of Fe catalysts, facilitating the formation of iron carbide phases, and inhibiting the over-hydrogenation process by strengthening the desorption of desirable products (especially alkenes). Consequently, Fe/MC1000 catalyst without Na shows lower selectivity for C2+ alcohols and higher selectivity for CH4 and paraffins.

Update

p.6. “The Fe-based component served as the active site, facilitating the dissociation of CO2, car-bon chain growth, and the insertion of oxygen-containing intermediates. Consequently, the Na/MC1000 sample was inactive due to the absence of these crucial sites (Table 2). Na acts as a promoter, increasing the surface basicity of Fe catalysts, facilitating the formation of iron carbide phases, and inhibiting the over-hydrogenation process by strengthening the desorption of desirable products (especially alkenes) [38-40]. As a result, Fe/MC1000 catalysts without Na doping exhibited lower C2+ alcohols selectivity and higher selectivity of CH4 and paraffins (Table 2).”

Table 2. Catalytic performance of the NaFe/MCx catalysts in CO2 hydrogenation

Catalyst

CO2

Conv. (%)

CO

Sel. (%)

Hydrocarbons

MeOH

C2+OH

C2+OH STY mg gcat-1 h-1

Yield

CH4

C2-4=

C5+=

Paraffin

NaFe/MC700

6.1

87.0

90.9

1.1

0.0

8.0

0.0

0.0

0.0

0.0

NaFe/MC800

17.5

78.2

31.2

7.6

1.5

33.6

6.6

19.5

15.2

0.7

NaFe/MC900

22.3

62.1

23.8

17.3

5.7

29.2

2.8

21.3

36.2

1.8

NaFe/MC1000

22.8

70.8

22.5

28.2

3.9

20.0

2.7

22.6

30.3

1.5

Fe/MC1000

18.4

87.05

36.8

30.3

1.3

31.2

0.1

4.5

0.8

0.8

Na/MC1000

1.6

-

-

-

-

-

-

-

-

-

Reaction conditions: 320 oC, 5 MPa (23.75% CO2, 71.23% H2, and 5.02% Ar), 15 mL min-1, GHSV=9000 mL gcat-1 h-1, 1 g quartz sand, and time on stream (TOS) = 24 h. Catalyst weight: 0.1 g.

Ref. 40 have been cited in the revised manuscript.

  1. Yang, Q., Vita, A. K., Kondratenko, S. A., Petrov, D. E.., Dmitry, E. D., Erisa, S., Henrik, L., Aleks, A., Ralph, K., Andrey, S. S., Alexandr, A. M., and Evgenii, V. K.. (2022). Identifying Performance Descriptors in CO2 Hydrogenation over Iron-Based Catalysts Promoted with Alkali Metals. Angew. Chem. Int. Ed. 61, e202116517.

Comments 7: After the reaction, the significant increase in particle size of iron species is due to aggregation or the formation of carbides affecting pore size. However, Figure 4b shows that the catalyst remains stable after 25 hours of reaction. Is there a difference in product selectivity due to carburization or pore blockage during the reaction process? The selectivity results of the product within 0-25 hours should be reflected in the main text.

Response 7:

The catalyst spends an induction period at the start of the reaction to undergo the phase change stabilization process. There is no doubt that the crystal phase evolution process could affect the product distribution as reflected by the gradually increased CO2 conversion and significantly changed active phase. It has been widely accepted that the unstable product distribution during the induction period is useless for the practical valuable chemicals synthesis. However, after the induction period, the product distribution remains stable due to the unchanged active site. In order to verify the stable product distribution after induction period, we prolonged our reaction test to 48 hours. By comparing the product distribution obtained at 24 h and 48 h, it is reasonable to believe the stability of the carbon supported catalyst.  Furthermore, it should be noted that due to the design of the testing equipment and the products analysis method, only the products collected over a long time could be analyzed correctly, and the amount of the hourly produced product is too small, resulting in a large analysis error.

Update

p.6. “NaFe/MC1000 exhibited the highest CO2 conversion (22.8%) and selectivity towards C2+OH (22.6%), alongside the lowest CH4 selectivity (22.5%). Furthermore, over a 48-hour testing period, the CO2 conversion and product distribution of NaFe/MC1000 remained stable without being impacted by carbon deposition or pore blockage (Table 2 and Figure 4c, d).”

p.6. "Figure 4. (a) Product distribution of the NaFe/MCx catalysts in CO2 hydrogenation. (b) CO2 conversion rate of NaFe/MCx over time. (TOS) = 24 h. (c) CO2 conversion rate and (d) product distribution of NaFe/MC1000 over time. (TOS) = 48 h.

Round 2

Reviewer 2 Report

Comments and Suggestions for Authors

The authors have been made revision to enhance quality of the manuscript, it could be published as it is.